**SOFTWARE**

# Producing polished prokaryotic pangenomes with the Panaroo pipeline

Gerry Tonkin-Hill[1,2*] ⓘD, Neil MacAlasdair[1,3], Christopher Ruis[3,4,5], Aaron Weimann[3,4,5,6] ⓘD, Gal Horesh[1], John A. Lees[7], Rebecca A. Gladstone[2], Stephanie Lo[1], Christopher Beaudoin[8], R. Andres Floto[4,9], Simon D.W. Frost[10,11], Jukka Corander[1,2,12†], Stephen D. Bentley[1†] and Julian Parkhill[3†]

*Correspondence: gt4@sanger.ac.uk
†Jukka Corander, Stephen D. Bentley and Julian Parkhill contributed equally to this work.
[1]Parasites and Microbes, Wellcome Sanger Institute, Cambridge, UK
[2]Department of Biostatistics, University of Oslo, 0317, Blindern, Norway
Full list of author information is available at the end of the article

**Abstract**

Population-level comparisons of prokaryotic genomes must take into account the substantial differences in gene content resulting from horizontal gene transfer, gene duplication and gene loss. However, the automated annotation of prokaryotic genomes is imperfect, and errors due to fragmented assemblies, contamination, diverse gene families and mis-assemblies accumulate over the population, leading to profound consequences when analysing the set of all genes found in a species. Here, we introduce Panaroo, a graph-based pangenome clustering tool that is able to account for many of the sources of error introduced during the annotation of prokaryotic genome assemblies. Panaroo is available at https://github.com/gtonkinhill/panaroo.

**Keywords:** Bacteria, Pangenome, Prokaryote, Clustering, Horizontal gene transfer

## Background

Prokaryotic genome evolution is driven both by the transfer of genetic material vertically from parent to offspring and by horizontal gene transfer between organisms [1]. Large population sequencing studies of bacteria have confirmed that this results in large-scale differences in intraspecies genome content [2]. This has led to the description of the pangenome, the set of all genes that have been found in a species as a whole [3]. Within the pangenome, genes are often then described as being part of the 'core' genome, the set of genes present in all members of a species, or the non-core ('accessory') genome. Throughout this paper, we refer to the problem of correctly identifying all the gene families that are present in a collection of annotated assemblies as both inferring and determining the pangenome.

A common problem when inferring the pangenome of bacterial genomic datasets is the classification of homologous genes, usually defined by a percentage shared sequence identity, into either orthologous or paralogous clusters. Orthologs trace their most recent common ancestor to a speciation event whereas paralogs trace their most recent common

ancestor to a gene duplication event. When analysing bacterial pangenomes, we are often interested in identifying paralogs as genes with near identical sequence may perform a different function or be under differential regulation at different locations in the genome. Many programmes for pangenome analysis therefore use location information to further identify paralogs, as well as xenologs which occur when gene duplications are acquired through horizontal gene transfer.

Previous approaches which aid in the inference of the pangenome of a collection of bacterial isolates include Roary, OrthoMCL, PanOCT, PIRATE, PanX, PGAP, COG-soft, MultiParanoid, PPanGGoLiN and MetaPGN [4–12]. The majority of methods for determining the pangenome tend to make use of one of two similar approaches (see Supplementary Figure 1). Most start by inferring similarity between predefined gene sequences using a homology search tool such as CD-HIT, BLAST or DIAMOND [13–15]. Using this output, a pairwise distance matrix is created and genes are then clustered into orthologous groups either using the popular Markov Clustering algorithm (MCL) or by looking at triangles of pairwise best hits (BeTs) [16, 17]. A subset of these methods then use gene adjaceny information to build a graphical representation of the pangenome. This graph is then used to further split orthologous clusters into paralogs. Roary, PIRATE, PPanGGoLiN and MetaPGN also provide this graphical representation as an output file. A final step of some pipelines is to classify the resulting clusters into core and accessory categories based upon their prevalence within the dataset. This is usually done using predefined thresholds; however, more recently model-based extensions to this approach have been suggested [11].

As bacterial genomic population studies have grown larger, there has not been a corresponding increase in genome annotation accuracy or genome assembly contiguity. Thus, as these databases have grown, so has the number of erroneous gene annotations. This can have profound implications for the resulting estimates of the number of gene families present, whereby a higher number genomes leads to a higher number of errors [18, 19]. Such errors can cause difficulties in any downstream modeling of the pangenome, such as the modeling of negative frequency-dependent selection (NFDS) acting through the loci in the accessory genome [20, 21]. Errors can be introduced into pangenome analyses by fragmented assemblies, mis-annotation, contamination and mis-assembly. Denton et al. have shown that fragmented assemblies were the major cause of inflated gene numbers in draft eukaryotic genomes [19]. Whilst errors often lead to inflations in the estimates of the size of the accessory genome, they can also lead to missing genes when the annotation software fails to identify a gene or where the gene is fragmented by a break in the assembly, which reduces the estimated size of the core genome. Many current pangenome inference algorithms have not been subjected to rigourous verification using simulated data. Consequently, their ability to deal with the errors occurring in the initial genome annotations has received limited attention.

Here, we present an alternative approach to inferring the pangenome, Panaroo, which makes use of a graph-based algorithm to share information between genomes, allowing us to correct for many of the sources of annotation error. Panaroo leverages the additional information provided by each genome in a dataset to improve annotation calls, and as a result, the clustering of orthologs and paralogs within the pangenome. We also provide a number of pre- and post-analysis scripts which further enrich the analysis package we provide, allowing integrated data quality control and gene association

analysis, and to allow for the comparison of pangenomes between species. Whilst these scripts can allow for comparisons of the resulting pangenomes between species, Panaroo is not recommended for metagenomic datasets. As Panaroo constructs a full graph representation of the pangenome, we are able to investigate structural variations within the resulting graph, allowing for associations between structural variations and phenotypes to be called. We demonstrate the success of the algorithm through extensive simulation using the Infinitely Many Genes model [22] and by analysing a diverse array of large bacterial genomic datasets including the major clades of the Global Pneumococcal Sequencing (GPS) project [23]. We compare the output of Panaroo with the previous gold standard methods for analysing the pangenome and show that Panaroo produces superior ortholog clusters, often leading to both significant reductions in the size of the estimated accessory genome and increases in the size of the core genome.

## Results

### Overview

Panaroo builds a full graphical representation of the pangenome, where nodes are clusters of orthologous genes (COGs) and two nodes are connected by an edge if they are adjacent on a contig in any sample from the population. Using this graphical representation, Panaroo corrects for errors introduced during annotation by collapsing diverse gene families, filtering contamination, merging fragmented gene segments and refinding missing genes (Fig. 1). Panaroo generates the initial gene clusters using CD-HIT to cluster the collection of all gene sequences in all samples [13]. Paralogs are then split by only allowing each genome to be present once in each cluster. Fragmented or mistranslated genes are identified and merged based on neighbourhood information of each node [24]. Diverse gene families are identified using a relaxed alignment threshold along with neighbourhood information obtained from the graph. Potentially contaminating genes with low contextual support in the graph are then optionally removed. This retains rare genes that have reliable contextual support. Finally, genes potentially missing from one or more samples are identified in the graph and the contig sequence near neighbouring nodes is searched to check for the presence of the gene.

Panaroo uses a number of predefined thresholds to construct the pangenome graph. These can all be adjusted by the user, but we provide a number of modes for common use cases. In the 'strict' mode, Panaroo takes a more aggressive approach to contamination and erroneous annotation removal. This is most useful when investigating genomes where rare plasmids are not expected or when phylogenetic parameters such as gene gain and loss rates are of interest. In these cases, erroneous gene clusters can quickly dominate the estimated parameters. In its 'sensitive' mode, Panaroo does not remove any gene clusters. This is useful if a researcher is interested in rare plasmids which may be hard to distinguish from contamination. When running Panaroo in sensitive mode, it is important to be aware of the possibility of a higher number of erroneous clusters. In the following analyses, we run Panaroo in both its 'strict' and 'sensitive' modes with Panaroo generally outperforming all other tools even without contamination removal in the 'sensitive' mode.

Panaroo takes annotated assemblies in GFF3 format as input and generates a variety of output formats including a gene presence absence matrix (as in Roary) as well as a fully annotated graph in GML format for viewing in Cytoscape or other graph visualisation software [25]. The Panaroo package includes a number of pre- and post-processing scripts

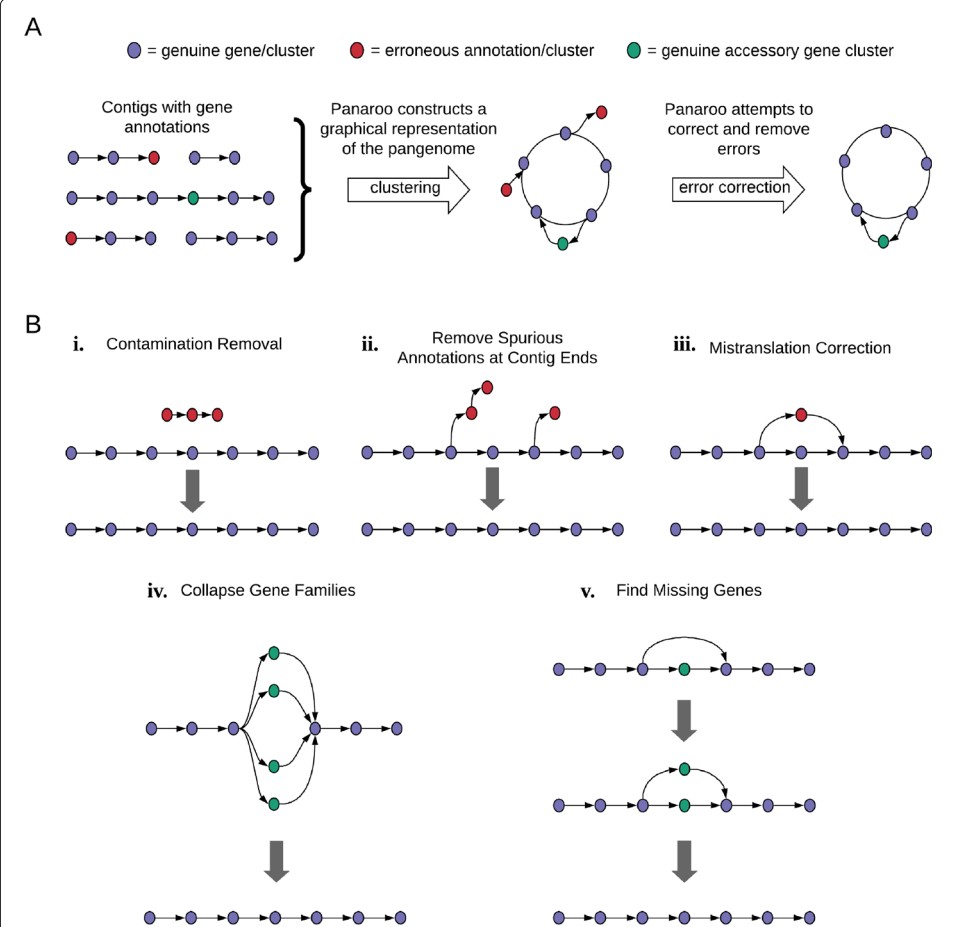

**Fig. 1 a** An overview conceptualising the problem with current gene annotation methods and the stages Panaroo uses to correct for annotation errors. **b** Expanded specific stages in the process. (i) Contamination appears in the graph as poorly supported components. In the default mode, Panaroo removes contamination by recursively removing poorly supported nodes of degree 1. (ii) Genes are often mis-annotated near contig breaks [19]. Panaroo corrects such mis-annotations by recursively removing poorly supported nodes of degree 1. (iii) Panaroo corrects cases where the same DNA sequence has been translated in multiple reading frames into a single gene by clustering concomitant genes at the DNA level. (iv) Panaroo uses context and a lower clustering threshold to combine diverse gene families into a single gene. (v) Annotation algorithms may predict a gene in some but not all samples, even when the samples share exactly the same DNA sequence. Panaroo finds missing genes by searching for the gene sequence in the surrounding DNA

that can be used for initial quality control as well as for determining pangenome size, gene gain and loss rates and to identify coincident genes. Panaroo interfaces easily with many other pangenome analysis packages including the latest version of pyseer allowing for associations between phenotypes and gene presence/absence as well as structural variation in the graph to be investigated [26]. The package is written in python and is available under an open source MIT licence from https://github.com/gtonkinhill/panaroo/.

### Corrected analysis of a *Mycobacterium tuberculosis* outbreak in London

To assess the effectiveness of Panaroo and the impact of annotation errors on other pangenome inference methods, we analysed a large outbreak of highly clonal, isoniazid-resistant *Mycobacterium tuberculosis* (Mtb) in London [27]. Mtb exhibits a very low mutation rate and is understood to have a 'closed' pangenome. Due to the short timescale

of the outbreak, the maximum pairwise SNP distance within this dataset was 9. As we would expect to find no pangenome variation, this dataset provides a useful control to compare the different pangenome tools.

We ran each of the pangenome inference methods on all 413 Mtb genome assemblies after first annotating them using Prokka [28]. Panaroo identified both the highest number of core genes and the smallest accessory genome (Fig. 2), consistent with the established biology of Mtb and a highly clonal dataset [29, 30]. In contrast, PanX, PIRATE, PPanG-GoLiN, COGsoft and Roary all reported inflated accessory genomes ranging in size from 2584 to 3670 genes representing a nearly tenfold increase to that reported by Panaroo. In its default mode PPanGGoLiN reported over 10,000 gene clusters giving it the highest error rate. However, this reduced to 7131 after enabling the –defrag parameter. We thus ran PPanGGoLiN using this parameter in all analyses. The small number of accessory genes that Panaroo did predict mostly consisted of core genes where the algorithm was unable to refind the genes in a subset of the assemblies. The majority of the difference between the methods was driven by genes being fragmented during assembly (59%; see Supplementary Methods). A smaller subset of genes were only called in a small minority of the isolates despite the underlying sequence being nearly identical (10%). Whilst some of these differences could be due to frame shifts in the PE/PPE genes, 27.9% of the isolates were indistinguishable with only one isolate being more than 5 SNPs from this major clone. We found that the majority of the difference was due to the annotation algorithm optimising for each isolate individually, leading to inconsistent gene calls. However, Panaroo's consensus approach helps to resolve these discrepancies. The magnitude of the difference observed in this dataset suggests that failing to account for annotation errors can have profound impacts on the resulting estimates of the pangenome.

An alternative to correcting gene annotations is to perform strict quality control checks on assemblies prior to running pangenome inference tools. For very highly contaminated

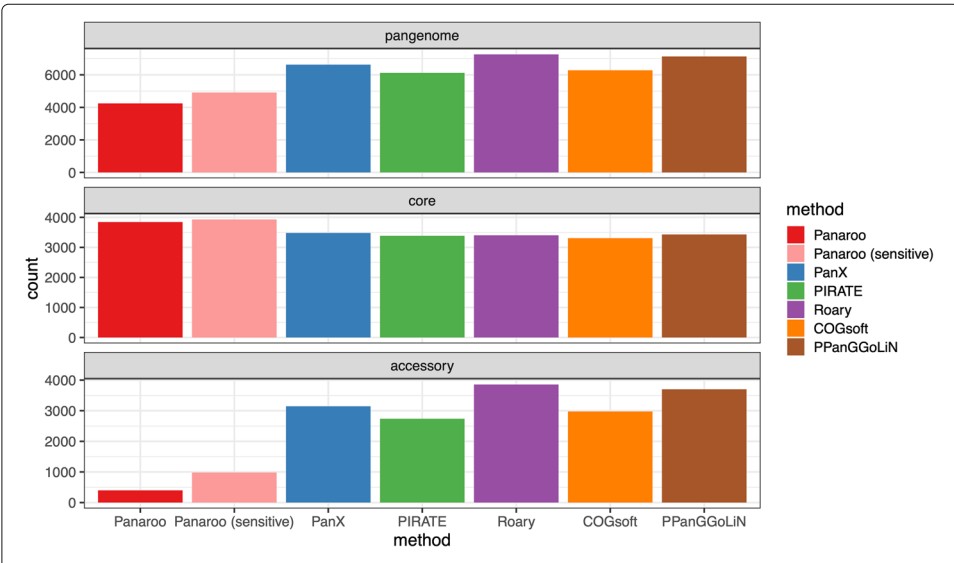

**Fig. 2** Pangenome counts for 413 *Mycobacterium tuberculosis* genomes from an outbreak in London [27]. The maximum pairwise SNP distance between these isolates was 9, suggesting extremely limited variation. Consequently, we would expect a very limited accessory genome and a core genome of approximately 4000 genes. All tools with the exception of Panaroo found in excess of 2500 accessory genes, which can be attributed to annotation errors

assemblies or those of very low quality, this can be the best option and Panaroo includes a quality control pre-processing script for this purpose. However, in many cases, low-level contamination and fragmented assemblies are common, and thus, filtering out assemblies with minor errors can lead to large data losses. In large collections, even very low annotation error rates will eventually compound pangenome inference results. To investigate such a strategy on the Mtb dataset, we ran CheckM, which is a common assembly quality control pipeline [31]. CheckM produces completeness and contamination scores by using a reference gene dataset to compare with assemblies. The resulting scores on the Mtb dataset are given in Supplementary Figure 2. As we know that this dataset should contain highly similar assemblies, it is possible to identify a number of problematic genomes with slightly lower completeness scores. If we were to remove these genomes, it would result in a loss of 12% of the dataset which could potentially have a large impact on downstream analysis. Instead, using Panaroo, we are able to retain these assemblies whilst controlling the error rate.

## Superior performance on simulated populations

To further assess the ability of the different methods to identify the correct gene presence/absence matrix, we simulated pangenomes using the *Escherichia coli* reference genome ASM584v2 (accession number NC000913) and the Infinitely Many Genes model [22, 32]. To more accurately simulate the kind of errors that typical annotation pipelines produce, we simulated short read assemblies from these pangenomes using Mason, ART and SPADES [33–35]. A more detailed description is given in the 'Methods' section. We conducted five simple and two more complicated simulations, each with three replicates (Supplementary Table 1). In the simple simulations, the gene gain/loss rate was varied with lower rates corresponding to a larger core genome and smaller accessory genome, whereas higher rates corresponded to a larger accessory genome. The mutation rate of the accessory genome was also varied. In addition, we simulated two more complicated datasets, one of which had an increased level of fragmentation of the assembly by fragmenting the input genome prior to the NGS simulation. This resulted in a mean N50 of 23,112. The second more complex simulation included contamination by randomly adding in short fragments of the *Staphylococcus epidermidis* reference genome, which is a common contaminant. The more complicated simulations represent datasets with unusually high error rates but help to clarify the large impact that these sources of error can have on pangenome inference as was previously demonstrated in the analysis of the highly clonal Mtb dataset.

Figure 3a indicates the number of gene clusters which contained errors for each of the scenarios. Such errors included genes that were missing, incorrectly annotated or had incorrectly clustered together. Most methods performed fairly well when applied to the output from the simple simulation. All methods include some errors due to genes never being annotated except in the original reference. As each method relied on the same input files, this was consistent between methods.

For the simple simulations, PanX and Panaroo produced the fewest errors, followed by PIRATE, Roary, PPanGGoLin and COGsoft. Roary was the most sensitive method to the substitution rate, with higher rates leading to more errors. This can be attributed to its reliance on a strict BLAST e-value threshold. COGsoft gave variable results, performing poorly on pangenomes with larger accessories suggesting it may over collapse genes. This

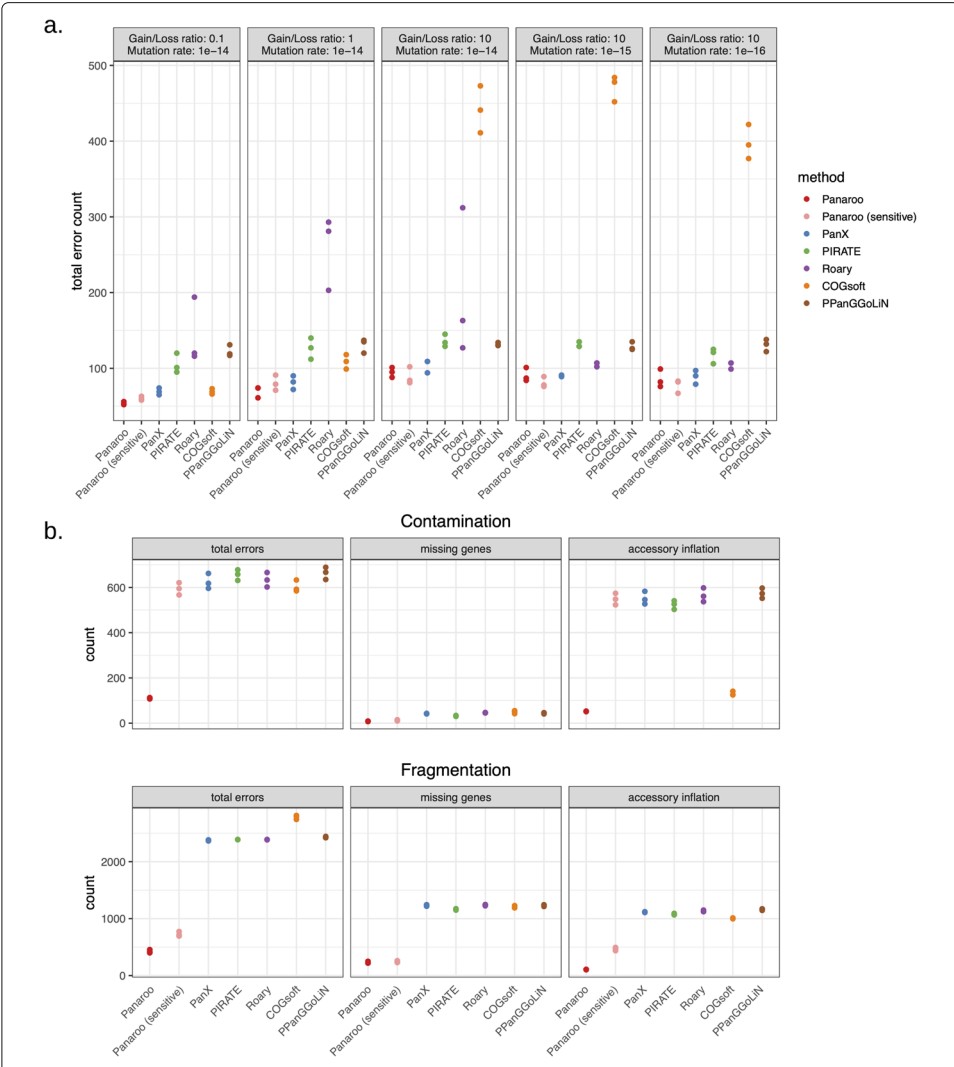

**Fig. 3** Error counts for the different algorithms after comparing with simulated data on different scenarios. Accessory genome inflation refers to the number of erroneous clusters that do not correspond to any simulated gene cluster. Missing genes refer to false-negative gene calls where the annotation is not present in the final pangenome. Even in simulations of pangenome variation from a single *E. coli* reference with only relatively simple sources of error, **a** Panaroo outperforms other methods across a variety of gene gain/loss rates and mutation rates. In more realistic simulations of sequencing data, **b** the only method with reasonable control of the error rate is Panaroo

interpretation was further supported in our analysis of a diverse *Klebsiella pneumoniae* dataset (see below).

Whilst most methods were able to perform adequately on relatively error-free simulated data, the introduction of more realistic significant sources of annotation error had a large impact. Figure 3b indicates the resulting error counts after simulating both contamination and highly fragmented assemblies. Here, the importance of Panaroo's multiple annotation error correction approaches becomes apparent. As expected, when small amounts of contaminating *S. epidermidis* DNA were added to the simulated NGS data, all methods except Panaroo and COGsoft incorrectly called a larger accessory genome. This is due to their inability to account for and remove contaminating contigs. In contrast, Panaroo achieved error rates similar to those found for the clean assemblies. As expected Panaroo's

sensitive mode did not correct for the additional contamination as potential contamination is not removed in this mode. COGsoft had a similar number of total errors to the other programmes but rather than calling a larger accessory genome tended to incorrectly merge the contamination with other genes.

The highly fragmented assemblies led to the largest error rates in each pangenome analysis tool. Fragmentation can lead to gene annotation software such as Prokka miscalling genes near the ends of contigs. It can also impact on the consistency of the training step in some annotation algorithms. This resulted in a large increase in the estimated accessory genome size for all methods except Panaroo. Similarly, miscalling can lead to genes being left unannotated resulting in smaller estimates of the core genome. In both cases, Panaroo's error correction and refinding steps were able to accurately recover the true pangenome, whilst PanX, COGsoft, PIRATE, PPanGGoLiN and Roary all produced nearly an order of magnitude higher error rates. Whilst Panaroo's sensitive mode produced far cleaner results than the other tools, the fact that it does not delete clusters prevents it from removing some spurious annotations. These results mirror that observed in the analysis of the highly clonal *M. tuberculosis* outbreak, helping to confirm the impact that such errors can have on pangenome estimates.

### Greater internal consistency in a diverse *Klebsiella pneumoniae* collection

We then went on to compare each method on a more complex real dataset—328 globally sourced *Klebsiella pneumoniae* genomes from both human and animal hosts [2]. *K. pneumoniae* is a highly diverse gram-negative bacterium that can colonise both plants and animals and has previously been found to have a large pangenome [2]. The high recombination rate and often multiple plasmids per bacterium complicates analysis of the *K. pneumoniae* pangenome. Nine of the 328 isolates were identified as outliers by the Panaroo quality control script due to the number of contigs or number of genes they contained (see Supplementary Figures 4-6). These isolates were removed before running each algorithm. Figure 4a indicates the resulting total, core and accessory gene counts inferred by each method, using the 99% presence threshold for core genes as used in Roary [4].

As species such as *K. pneumoniae* are known to have many rare plasmids which are difficult to distinguish from contamination, Panaroo's sensitive mode is of particular relevance here. Panaroo identified the highest number of core genes in both its default and sensitive modes, 3372 and 3376 respectively. Hence, for these genomes, there was only a minor difference in the estimated core between the two pipeline options. Roary identified the smallest core genome of 1800 genes. Given the result of the simulations, this is likely due to gene clusters being incorrectly split into multiple smaller clusters, as the default Roary pairwise identity threshold of 95% is too stringent for such a diverse dataset. PIRATE relaxes the strict threshold required in Roary, and it identified a similar number of core genes to Panaroo (3318) but a smaller number of accessory genes than both the Panaroo (sensitive) and PanX methods which agreed more closely with the original estimates in Holt et al. [2].

Whilst there is no gold standard with which to compare these results, we can look at the gene annotations within clusters to identify cases where a gene cluster contains multiple different annotations, which would suggest separate gene clusters have been incorrectly merged. Figure 4b indicates the number of conflicting annotations in the clusters of each method. As gene fragments and genes annotated as 'hypothetical' are often the result

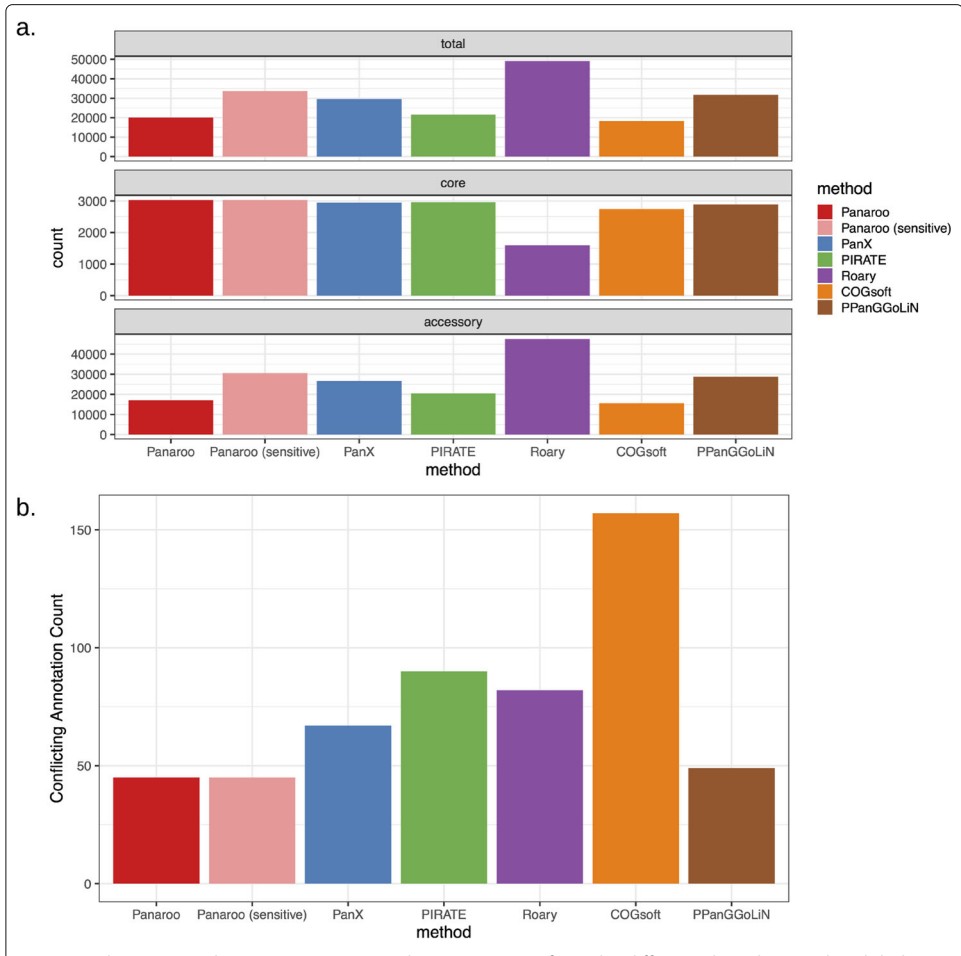

**Fig. 4 a** The estimated pangenome, core and accessory sizes from the different algorithms in the global *K. pneumoniae* dataset. **b** The number of conflicting gene annotations in the inferred clusters of the different algorithms

of errors and thus can have erroneous annotations, we did not consider conflicts that involved these. Panaroo in both its default and sensitive modes had the lowest number of conflicting annotations. PPanGGoLiN had the second lowest number, whilst COGsoft recorded the highest number of conflicts which is consistent with the tendency of its method to overcluster genes. The lower number of conflicting annotations found by PPanGGoLiN is consistent with it favouring splitting gene clusters over merging them as we found earlier. Overall, Panaroo identified a larger core genome and fewer conflicting annotations than any other method showing that its error correction approach is also suitable for diverse datasets of highly recombinogenic bacteria.

### Pyseer association analysis with Panaroo finds antibiotic resistance mechanisms

Panaroo provides a number of outputs as well as post-processing scripts for analysing the cleaned pangenome graph. Panaroo outputs both a gene presence/absence matrix as well as structural variation presence/absence matrix that can be used as input to pyseer or Scoary for association analyses [26, 36]. Panaroo generates structural variation calls by identifying distinct consecutive triplets of gene families in the graph that describes different paths through a node (see Fig. 5a). As larger insertion and deletion events will only

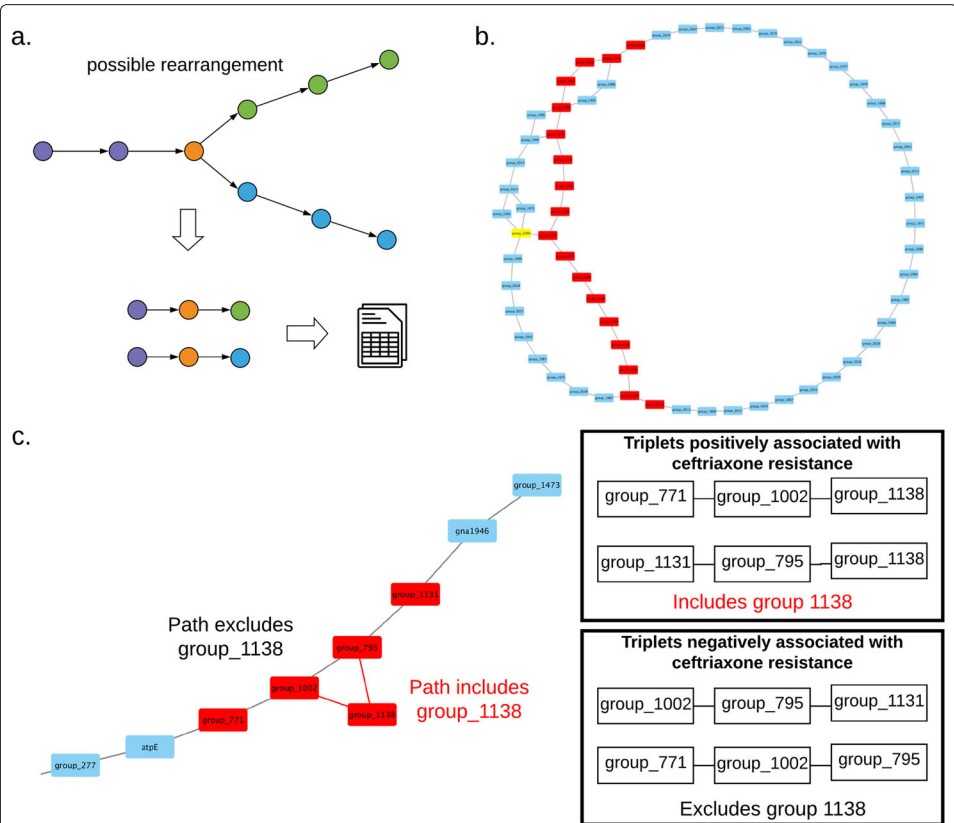

**Fig. 5 a** A diagram indicating how gene triplets are called in the graph. A single genome can only pass through a node once; thus, variations in the arrangement of genes in different genomes can be called using triplets. These triplets are summarised as a binary presence/absence matrix. **b** A family of related plasmids present in the *N. gonorrhoeae* pangenome gene network. The path highlighted in red contained 4 structural variant gene triplets significantly negatively associated with tetracycline resistance, or associated with tetracycline susceptibility by a structural variant pan-GWAS (all adjusted *p* value < 0.05 ). The gene highlighted in yellow, group_1999, was found to be a tetM resistance gene. **c** A subsection of the *N. gonorrhoeae* pangenome gene network of the region surrounding gene group_1138. The presence of gene triplets (group_771-group_1002-group_1138) and (group_1131-group_795-group_1138) is positively associated with tetracycline resistance whilst the triplets (group_1002-group_795-group_1131) and (group_771-group_1002-group_795) are negatively associated with tetracycline resistance (all adjusted *p* value < 0.05)

be represented once in the structural presence/absence matrix rather than repeatedly for each gene, this approach increases the power of such association analyses. The approach also identifies associations with large structural rearrangements although these are often more difficult to interpret. Once a significant association between a gene triplet and a phenotype of interest have been identified, the context of the structural rearrangement can be investigated manually by interrogating the pangenome graph in Cytoscape [25]. Panaroo is only able to call large structural rearrangements that result in genes being relocated within the genome. Finer scale structural variants are better called using assembly graph-based approaches such as Cortex [37].

To validate the pan-genome-wide association study (pan-GWAS) and pan-genome structural variant association study (sv-pan-GWAS) pipelines, we ran Panaroo on the Euro-GASP collection of 1054 *Neisseria gonorrhoeae* isolates collected from 20 countries

across Europe from September to November 2013 [38]. We combined the Panaroo output with antimicrobial MIC testing results for seven different antibiotics performed in the original study and carried out association studies on the gene presence/absence patterns and structural variants using pyseer [26].

The gene presence/absence pan-GWAS approach returned 67 genes (Supplementary Table 2) associated with various antibiotics (adjusted $p$ value $\leq 0.05$ ). This included many probable candidates for genes causing resistance, including an uncharacterised ABC transporter (group_464), for penicillin resistance. ABC transporters are a common resistance mechanism against ribosome-targeting antimicrobials, as they can function as efflux pumps [39].

The structural variant pan-GWAS returned 138 triplets (Supplementary Table 3) associated with antibiotic resistance (adjusted $p$ value $\leq 0.05$). These included many triplets containing phage-associated, transposase, or pilin genes, all of which are known to be mobile within the genome.

Among these, the sv-pan-GWAS identified a number of insertions and deletions of whole genes which were associated with antibiotic resistance. One of these, group_1138, is a transmembrane protein which, when inserted, is associated with ceftriaxone resistance. All four possible gene triplets bypassing or going through the insertion were significantly associated with either susceptibility or resistance depending on if they included group_1138. The mechanisms of ceftriaxone resistance in *N. gonorrhoeae* are not yet fully understood, but it has been suggested that efflux and permeability must play a role [40]. Group_1138, as it is a transmembrane protein, could have either of these functions.

The sv-pan-GWAS approach allows for closely related genetic architectures to be disentangled, including highly related plasmids and phages. For example, analysis of the pangenome graph showed that a common *N. gonorrhoeae* plasmid present 430 times in this dataset is actually a family of several closely related plasmids. These highly similar plasmids share the majority of their genes, but there are several differences in gene content, which appear as bubbles in the pangenome graph (Fig. 5b). One of the plasmid versions (highlighted in red in Fig. 5b) is negatively associated with tetracycline resistance, with four gene triplets significantly negatively associated with this phenotype in the sv-pan-GWAS. The other plasmid variants each contain group_1999, a tetM tetracycline resistance gene, providing a mechanism to explain the differential resistance profiles. Together, these analyses demonstrate that multiple members of the same plasmid family with different resistance profiles are circulating in the European *N. gonorrhoeae* population, and illustrate the value of an the sv-pan-GWAS approach.

**Improved methods for analysing pangenome evolutionary dynamics**

The higher accuracy obtained by Panaroo allows for the comparison of gene gain and loss rates between lineages and species as well as the more accurate inference of pangenome size. Whilst it is a common practice to plot gene accumulation curves in the analysis of pangenomes, these are not robust to errors and fail to account for sampling biases and population structure. Thus, accumulation curves should not be used to compare pangenome characteristics of different lineages or species. Recently, a number of phylogenetically informed methods for investigating pangenome dynamics have been published, including the Infinitely Many Genes (IMG) model and the Finitely Many Genes (FMG)

model [8, 22, 41]. Both of these approaches account for the diversity of the sample and have been implemented as post-processing scripts in Panaroo.

To demonstrate the utility of using the corrected pangenome graph to infer gene gain/loss rates and pangenome size, we used the FMG model to investigate 51 of the major Global Pneumococcal Sequence Clusters (GPSCs) for which reliable dated phylogenies could be constructed [42]. The major clades of the pneumococcus have distinct accessory gene profiles [43]. We ran Panaroo on each GPSC separately and used the resulting gene presence/absence matrix with the corresponding dated phylogeny to infer gene gain and loss rates for each cluster. We compared the inferred parameters with other variables of interest calculated by Gladstone et al. [23], including the inferred recombination rate (r/m), odds ratio of invasive disease and the number of distinct serotypes for each cluster. The parameters along with these variables are plotted in Supplementary Figure 3. We found that the estimated effective pangenome size correlated positively with the recombination rate of a cluster (Spearman correlation coefficient 0.53, $p < 0.001$) and the number of serotypes present in the cluster (Spearman coefficient 0.51, $p = 0.001$). This is consistent with biological understanding of the genome diversification and gives confidence to our results, as a higher recombination rate would allow for a clade to more easily gain and lose genes, including serotype-defining gene clusters, resulting in a larger pangenome. Interestingly, GPSCs that have lower gene gain rates were more likely to have a significant odds ratio for invasive disease ($p = 0.04$)(see Fig. 6). The association with gene loss rate was weaker, although the effect was in the same direction ($p = 0.08$). Genome reduction has previously been associated with increasingly obligate interactions with the host in multiple unrelated bacterial pathogens [44].

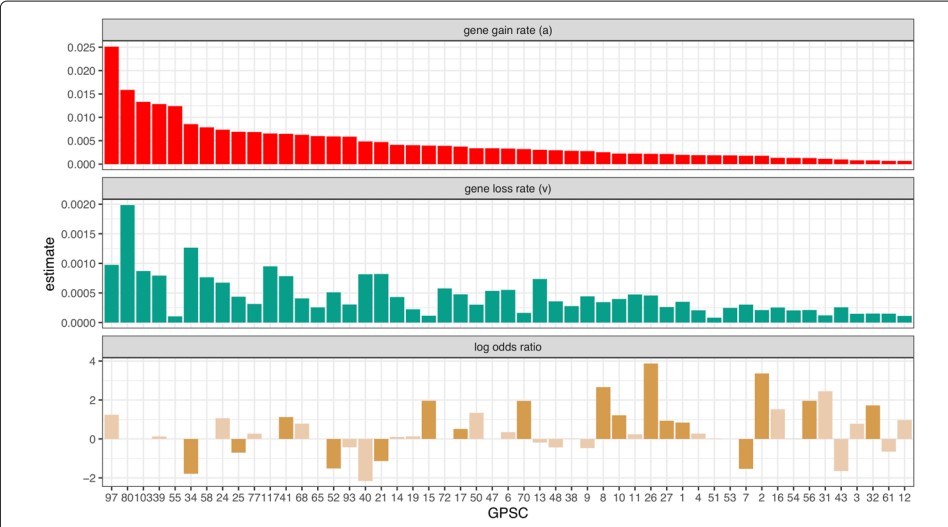

**Fig. 6** The inferred gene gain and loss rates of each of the 51 major clades of the Global Pneumococcal Sequencing project plotted above the respective log odds ratio of invasive disease in that clade. Clades which had significant odds ratios in Gladstone et al. [23] are represented in dark yellow

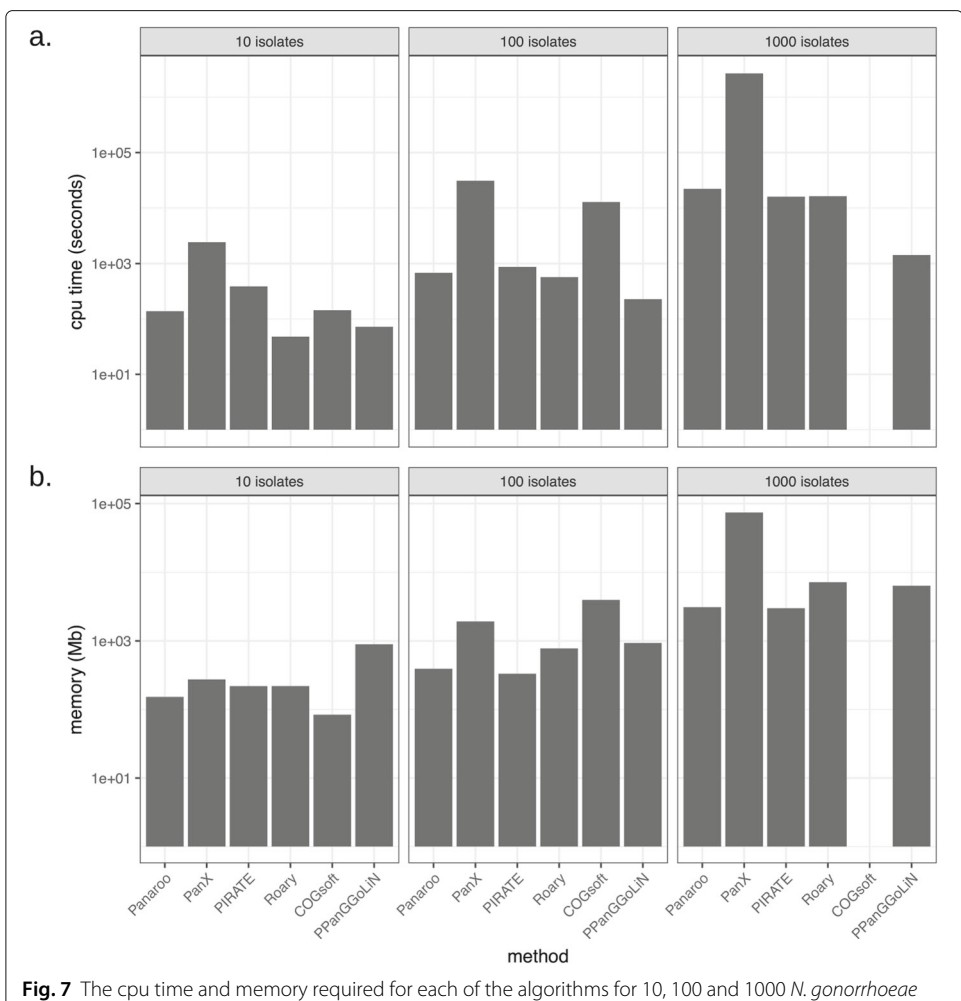

**Fig. 7** The cpu time and memory required for each of the algorithms for 10, 100 and 1000 *N. gonorrhoeae* isolates. Each tool was run with 5 cpus

## Computational performance

Panaroo uses a similar level of computational resources to competing methods. Figure 7 indicates the memory and cpu time required for the analysis of 10, 100 and 1000 *N. gonorrhoeae* isolates subsampled from the Euro-GASP collection. PanX and COGsoft used the most resources with COGsoft not completing the largest dataset in under a week. Roary, PIRATE and Panaroo all performed similarly.

## Discussion

Annotation errors, fragmented assemblies and contamination represent a major challenge for pangenome analysis. We have designed Panaroo to tackle these challenges using a sophisticated framework for error correction that leverages information across strains through a population graph-based pangenome representation. Using both simulations and well-characterised real-world datasets, we demonstrated that many commonly used methods greatly inflated the size of the accessory genome whilst reducing the estimated size of the core genome. In contrast, Panaroo exhibited far lower error rates and reconstructed highly accurate core and accessory genomes for simulated datasets that included contamination and genome fragmentation. Analysis of both a highly conserved

*M. tuberculosis* dataset and a highly diverse *K. pneumoniae* dataset indicated that Panaroo provides superior solutions in challenging real-world population genomics applications.

Panaroo also includes a number of pre- and post-processing scripts for the analysis of bacterial pangenomes that assist in quality control of the input data and facilitate downstream processing of the pangenome. We used the Panaroo pre-processing QC scripts to identify nine *K. pneumoniae* samples that were outliers based on the number of contigs or genes and excluded these samples from our analysis. We recommend that such pre-processing QC be carried out on all datasets to identify potentially erroneous samples.

We used the output from Panaroo as input to pyseer to run pan-GWAS and sv-pan-GWAS analyses on *N. gonorrhoeae*. Through this approach, we identified a deletion in the genome of *N. gonorrhoeae* in a large European collection that confers resistance to tetracycline. We demonstrated the utility of Panaroo to disentangle highly similar genetic structures through identification of a plasmid family in *N. gonorrhoeae* (Fig. 5c). By combining this high-resolution picture with structural variant pan-GWAS, we identified that some members of this plasmid family carry tetracycline resistance and were able to accurately determine the tetM gene as the cause of resistance.

As part of the Panaroo package, we include implementations of recently proposed pangenome evolution models, which are more appropriate than the more frequently used gene accumulation curves. We demonstrated the effectiveness of such methods through the analysis of the 51 major GPSCs where we observed an association between recombination rate and pangenome size (Supplementary Figure 2). We also identified an association between pneumococcal clade invasiveness and gene gain rate.

Panaroo is written in python (versions 3.6+) and is available under the open source MIT licence from https://github.com/gtonkinhill/panaroo. The code used to produce the analyses described above along with summary data is available from https://github.com/gtonkinhill/panaroo_manuscript. The raw GFF3, FASTA and all intermediate post-processing files are available from https://doi.org/10.5281/zenodo.3599800. Taking gene annotation errors into account is vital to recover an accurate pangenome, something previous methods have struggled to do in a systematic manner. Panaroo uses gene adjacency in a population-graph to provide a fast method for pangenome analysis, which is robust to a wide range of error sources. In the future, we plan to further improve the computational performance of Panaroo to allow it to scale to datasets involving hundreds of thousands or millions of genomes. We will also extend the post-processing tools available to analyse the resulting pangenome graph.

## Methods

### Panaroo algorithm

The Panaroo algorithm builds a graphical representation of the pangenome where nodes are genes and edges connect nodes if two genes appear adjacent to one another on at least one contig. The algorithm then uses this initial graph structure to perform a number of cleaning steps which correct for many of the problems encountered in genome annotation. Panaroo accepts annotated assemblies in GFF3 format as output by the popular annotation pipeline Prokka [28]. Unlike similar pangenome software, Panaroo attempts to preserve the full global context of each gene in the graph. This is in contrast to other

programmes such as Roary [4, 7, 10] which uses only the local context surrounding genes to build the graph.

### Initial graph creation

To first build the graph, Panaroo runs CD-HIT (v4.8.1) at a high sequence identity threshold (98%) [13]. The resulting clusters are then either classified as non-paralogous gene clusters, if they contain at most one instance of each genome, or paralogous clusters if they contain more than one gene from any single genome. Initially, non-paralogous gene clusters are represented by a single node in the graph whilst paralogous clusters are split into a single node for every occurrence of that cluster in the dataset. For instance, if a paralogous gene appears twice in two genomes and once in another, the initial graph will contain five nodes representing that paralog. The graph is then built by connecting cluster nodes with edges between them if the two clusters appear adjacent to one another on any contig. Paralogous nodes are collapsed back into the maximum number of nodes in which those genes appear in a single genome using the global context of the graph. In the above example, this would result in the final graph having two instances of the paralog node.

### Contig ends

Fragmented assemblies can cause issues for gene annotation software, whereby genes are often mis-annotated near contig breaks [19]. These spurious annotations appear as short paths of low support edges and nodes that end in a node of degree 1 that splits off from the main graph. To deal with this, Panaroo recursively removes nodes of degree 1 that are below a given support threshold as indicated in Fig. 1.

### Contamination

Contigs originating from sample contamination are generally significantly diverged from the target species pangenome. Thus, contaminating contigs tend to appear as disconnected components from the main graph with low support. To remove these, Panaroo uses the same approach as described for contig ends to recursively delete low supported nodes with less than or equal to one degree (see Fig. 1). This approach has the advantage of retaining rare genes which are present in the main graph whilst removing likely contaminants. Whilst this has in general been found to be very successful, it can occasionally lead to rare plasmids being removed. We have found that the benefits of removing unwanted noise far exceed the small loss in sensitivity that this approach provides. However, we also provide three settings for the algorithm with the most sensitive retaining such rare calls which can be useful when one is interested in rare plasmids.

### Mistranslation correction

Many annotation algorithms rely on an initial training phase where their parameters are adapted to the dataset at hand [45–47]. Often, this training is performed separately on each genome. This is the case in the Prokka pipeline, which makes use of Prodigal to perform the initial gene annotation [28, 45]. This can result in an identical sequence being annotated differently in different genomes. To correct for this, Panaroo checks genes that are within close proximity in the pangenome graph to determine if any are likely to be mistranslations, frame shift mutations or pseudogenised gene copies

by comparing their sequence at the nucleotide level. If two gene sequence matches at a high coverage and identity, typically 95% and 99% respectively, a mistranslation is called and the gene node with the lower support is collapsed into the node with higher support.

### Collapsing gene families

Gene families diversify at different rates due to the influence of positive and purifying selection. This makes choosing a strict sequence identity threshold for defining orthologous clusters difficult. Most pangenome analysis software rely on either a pairwise sequence identity or BLAST e-value threshold. This reliance can lead to both overclustering, where separate gene families are incorrectly merged, and oversplitting where a single gene family is incorrectly split into several smaller clusters. Many approaches attempt to deal with the former of these problems by utilising contextual information to split apart clusters that have different gene neighbourhoods [4, 6]. More recently, alternatives that make use of clustering at lower thresholds followed by more involved splitting techniques have been proposed [7, 8]. As an alternative to these approaches, we extend the idea of using gene context to the oversplitting problem. Panaroo utilises gene contextual information to collapse diverse gene families that have been incorrectly split into multiple clusters during the initial pangenome graph creation. Initial gene clusters that share a common neighbour in the graph are compared at a lower pairwise sequence threshold (default 70%). If they fall within this threshold, the two nodes are collapsed and the resulting node is annotated to indicate it consists of a more diverse family. We have found that utilising this additional contextual information leads to more robust clusters.

### Identifying missing genes

Previous pangenome clustering software tools are unable to identify missing annotations. Gene annotations can be lost due to variability in model training, fragmented assemblies and mis-assemblies. Panaroo remedies this issue by identifying pairs of nodes in the pangenome graph where one node is present in a genome and its neighbour is not. The potentially missing node is then searched for in the sequence surrounding the neighbouring node. If a match of sufficient coverage and identity is found, the graph is corrected to include an annotation for this missing gene in that genome. The alignment tool edlib (v1.3.4) is used to perform these searches which enables millions of checks in a reasonable time frame [24].

### Output

To allow for simple integration with existing bioinformatics pipelines, Panaroo outputs many of the same file formats as Roary. This includes the same gene presence/absence file format as well as core and accessory genome alignments created using either MAFFT, Prank or Clustal Omega [48–50]. In addition, Panaroo outputs a fully annotated pangenome graph in GML format for easy viewing in Cytoscape [25]. Each gene node and edge is annotated with the genomes to which it belongs as well as the gene annotations given by Prokka, gene sequence and whether or not the node has been classified as being a paralog. This graph format provides a valuable tool for visually inspecting the results of Panaroo. As Panaroo attempts to build the full pangenome graph rather than only using

local context, this graph is able to provide insights hidden in many of the outputs of similar tools such as Roary [4].

### Structural variation

As Panaroo constructs the full pangenome graph, it is possible to go beyond gene presence/absence and look at the underlying structure of the graph. To facilitate the analysis of this structure, Panaroo generates a gene triplet presence/absence matrix, indicating when three genes are present in a path along a genome. This is demonstrated in Fig. 5a, and the resulting presence/absence matrix can be used in association studies to investigate differences in rearrangements between genomes in a species. The context of each triplet can then be analysed by looking at the full graph in Cytoscape.

### Pre- and post-processing

The Panaroo pipeline comes packaged with a number of pre- and post-processing scripts for analysing pangenomes. We have included a wrapper for the popular Mash and Mash screen algorithms which generates diagnostic plots for quality control prior to running the Panaroo pipeline [51, 52]. These plots include a multidimensional scaling (MDS) projection of pairwise Mash distances, interactive bar charts to investigate contamination, as well as gene and contig counts.

In addition, we have included post-processing scripts for estimating gene gain and loss rates using both the infinitely many genes (IMG) model [41, 53] and the finitely many genes (FMG) model of [8, 53]. These are preferable to the common practice of plotting accumulation curves to indicate pangenome size as they account for the diversity and timescale of the sampled isolates. This allows for a clearer comparison between the pangenomes of different species or clades. Panaroo also includes an implementation of the Spydrpick algorithm which allows for the identification of gene presence/absence patterns that are either highly correlated or anti-correlated whilst accounting for population structure [54]. Such correlations can indicate that the genes involved have epistatic effects on fitness or that their presence or absence is a result of similar selective pressures. Finally, the output of Panaroo seamlessly interfaces with pyseer (v1.3.0), a bacterial GWAS package [26, 55]. pyseer includes a wide range of methods for performing association studies allowing for phenotypic associations to be found with gene or structural presence/absence patterns.

### Simulation and comparison with previous methods

Using the *E. coli* reference genome ASM584v2 as a starting point, we simulated variation in the accessory genome by varying the rates of gene gain and loss using a phylogeny simulated with the Kingman coalescent in dendropy (v4.4.0) [56]. In addition, we simulated various degrees of sequence variation by varying the within gene codon substitution rate. Three replicate datasets of 100 sampled genomes were created for each set of model parameters outlined in Supplementary Table 1. Realistic sequence assemblies were generated by first simulating NGS sequencing reads using either Mason (v2.0.9) or ART (v2.5.8) [33, 35]. These were assembled using SPAdes (v3.13.0) [34]. The resulting assemblies were annotated using Prokka (v1.13.3) with a custom BLAST database containing the correctly assigned proteins from the simulation prior to assembly. This extensive simulation

pipeline provided more realistic data and included many of the sources of error encountered in pangenome analyses. To simulate the problems that fragmentation can bring to the analysis of pangenomes, we also simulated a fragmented assembly by breaking the simulated whole genomes into fragments prior to simulating the NGS reads. This resulted in highly fragmented final assemblies. Contamination was also simulated by randomly adding 10-kb segments of the *S. epidermidis* reference genome ASM764v1 a common lab contaminant to the simulated genomes prior to NGS simulation. These segments were added by sampling from a Poisson distribution with mean 1. The gene presence/absence matrix was then generated for PanX (v1.5.1), Roary (v1.007002), PIRATE (v1.0), COGsoft (v201204) and Panaroo (v1.0.0). These were compared with the simulated matrix and the number of inferred orthologous clusters that contained an error was counted and is shown in Fig. 3.

## Supplementary information

---

**Additional file 1:** Additional file 1 includes all supplementary tables and figures.

**Additional file 2:** Review history.

---

### Peer review information

### Acknowledgements

Many thanks to Lauren Bell for designing the Panaroo logo and to the members of teams 81 and 284 at the Wellcome Sanger Institute for helpful comments and testing.

### Review history

The review history is available as Additional file 2.

### Authors' contributions

GTH, NM, CR and AW conceived the project. GTH, NM, CR, AW, GH and JAL developed the method and software. GTH, NM, CR, AW, RAG, SL and CB performed the data analysis. RAF, SDWF, JC, SDB and JP provided supervision and aided in the interpretation of results. GTH, NM, CR and AW wrote the paper with contributions from all authors. The authors read and approved the final manuscript.

### Funding

Wellcome Trust [206194 to S.D.B, 107032/Z/15/Z to R.A.F]; Wellcome Trust PhD Scholarship Grant [204016 to G.T.H]; ERC [742158 to J.C.]. J.A.L. is funded by MR/R015600/1. This award is jointly funded by the UK Medical Research Council (MRC) and the UK Department for International Development (DFID) under the MRC/DFID Concordat agreement and is also part of the EDCTP2 programme supported by the European Union. C.R. is funded by the Botnar Foundation Research Award (6063), the UK Cystic Fibrosis Trust Innovation Hub Award (IH001).

### Availability of data and materials

The source code is freely available under the MIT licence at https://github.com/gtonkinhill/panaroo [57] and zenodo (https://doi.org/10.5281/zenodo.389445) [58]. The code for reproducing the figures is available at https://github.com/gtonkinhill/panaroo_manuscript. Archived data for replication at time of publication to bioRxiv is available at https://doi.org/10.5281/zenodo.3599800.

### Ethics approval and consent to participate

Not applicable.

### Competing interests

The authors declare that they have no conflict of interest.

### Author details

[1] Parasites and Microbes, Wellcome Sanger Institute, Cambridge, UK. [2] Department of Biostatistics, University of Oslo, 0317, Blindern, Norway. [3] Department of Veterinary Medicine, University of Cambridge, Cambridge, UK. [4] Molecular Immunity Unit, Department of Medicine, University of Cambridge, Cambridge, UK. [5] Medical Research Council (MRC)—Laboratory of Molecular Biology, Cambridge, UK. [6] European Bioinformatics Institute, Cambridge, UK. [7] MRC Centre for Global Infectious Disease Analysis, Department of Infectious Disease Epidemiology, Imperial College London, London W2 1PG, UK. [8] Department of Biochemistry, University of Cambridge, Cambridge, UK. [9] Cambridge Centre for Lung Infection, Royal Papworth Hospital, Cambridge CB23 3RE, UK. [10] Microsoft Research, Redmond, WA 98052 USA.

[11]London School of Hygiene & Tropical Medicine, London, UK. [12]Helsinki Institute for Information Technology HIIT, Department of Mathematics and Statistics, University of Helsinki, 00014 Helsinki, Finland.

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

## 
