## [**Additional file 2** Review history. · Genome Biology]

Review History

First round of review

Reviewer 1

Were you able to assess all statistics in the manuscript, including the appropriateness of statistical tests used? There are no statistics in the manuscript.

Were you able to directly test the methods? Yes.

Comments to author:

Please consider the following questions:

-Is the Software likely to be of broad utility? Is it easy to install and use?

It is of broad utility. My lab have installed it and have not reported any problems.

-Does the software represent a significant advance over previously published tools (as demonstrated by direct comparison with available related software)?

Yes. it is a lot better than its predecessors

-Please indicate briefly the novel features and/or advantages of the software and/or please reference the relevant publications and which alternatives, if any, it should be compared with.

There are good comparisons with ROARY, but also the authors do a good job in referencing and listing other relevant software. Perhaps even they have included programs that are not exactly relevant, but in the interests of completeness, they have included them.

-Are the methods appropriate to the aims of the study, are they well described, and are necessary controls included?

Not quite relevant. the authors analyse some datasets and have used the appropriate approaches.

-Are the conclusions adequately supported by the data shown?

N/A

-Are sufficient details provided to allow replication and comparison with related analyses that may have been performed?

Yes.

-Is the paper of broad interest to others in the field, or of outstanding interest to a broad audience of biologists?

I think it should be of interest to a lot of biologists and I have no doubt that this will be a highly-cited paper.

I have some other comments below:

Minor point:

This is by far the most awkward description of what an ortholog might be that I have ever seen: "Orthologs are homologous genes descended from the same ancestral sequence in the common ancestor, and not via gene duplication or acquisition."

Quite simply, Orthologs trace their most recent common ancestor to a speciation event. Paralogs trace their most recent common ancestor to a gene duplication event.

Minor point:

This is a somewhat strange statement: "When analysing bacterial pangenomes we are often interested in not just the function of a gene or protein but also its location, as two nearly identical genes could be under differential regulation at different locations in the genome."

Genes could be regulated differently and still be side-by-side in the genome. Their location and their regulation are orthogonal issues. Location does not necessarily connote regulation.

Minor point:

This is not quite correct: "Many programs for pangenome analysis therefore use location information to further identify paralogs, which occur when two genes descend from the same ancestral sequence due to gene duplication or when a homolog has been acquired horizontally."

When genes are in the same genome as a consequence of HGT, then they are called Xenologs. You can, in fact, have two orthologs in the same genome, if one of them was acquired by HGT. Your statement is conflating a few different issues and so I recommend you straighten out these issues before publication.

Minor point:

When you say "Previous approaches for inferring the pangenome include Roary, OrthoMCL, PanOCT, PIRATE, PanX, PGAP, COG-soft, and MultiParanoid (4-10). The majority of methods for determining the pangenome tend to make use of one of two similar approaches (see Supplementary Figure 1)." it is not clear what you mean by "inferring the pangenome" and "determining the pangenome". I suggest you define what you mean in these cases. Do you mean that these approaches can be used in order to help identify all the gene families that are present in a complete pangenome? On its own, a program like OrthoMCL doesn't specifically work on delineating the gene content of a pangenome. If you use it in a particular way, then it can help you carry out this task.

Minor point:

This sentence needs a bit of work: "This can have profound implications for the resulting estimates of the pangenome whereby a higher number of genomes leads to a higher number of errors (16, 17)."

Here you speak about "estimates of the pangenome". Again, I believe you are talking about estimating the complete set of gene families in a pangenome, along with their pattern of presence and absence.

Reviewer 2

Were you able to assess all statistics in the manuscript, including the appropriateness of statistical tests used? No, I did not feel adequately qualified to address the statistics.

Were you able to directly test the methods? Yes.

Comments to author:

Building pan-genomes is an important step to analyze strains within a microbial species at population scale. Accurate construction of pan-genomes (i.e. identifying core and accessory genes) in a species can be hampered by poor-assemblies (e.g., high contamination, fragmentation, or mis-assemblies). This work proposes a graph-based pan-genome clustering tool Panaroo that is partly immune to faults in genome assemblies and their annotation. Improvements over state-of-the-art tools to build pan-genomes are characterized using extensive simulations. In particular, authors claim that Panaroo avoids inflation of

accessory genome and underestimation of core genome caused by erroneous genome annotations. This topic is certainly of broad interest in bioinformatics. The paper is clearly written and easy to follow. Panaroo is also open-source, already being used and well-documented on GitHub.

MAJOR CONCERNS:

As many pan-genome construction tools are now available, text in the introduction and results section does not highlight the methodological advance. The use of graph-based representation for characterizing pan-genome is not novel. Pan-genome construction tools like PPanGGOLiN, MetaPGN and PIRATE also use similar graphs where genes are represented as nodes, and edges reflect gene adjacencies. In particular, Panaroo appears very similar to PPanGGOLiN. These tools should not only be discussed, but also included in comparison. In addition, graph-based representations have been leveraged before for detecting structural variants, e.g., see Cortex tool (Iqbal et al. 2012).

A novel aspect of Panaroo is detection of contamination using poorly supported regions in the graph. However, I did not see a convincing strategy to differentiate errors vs. rare genes or recombinations. From a practical perspective, it's unclear whether discarding poorly supported regions is a good idea as it also affects sensitivity of the method. Perhaps a better way could have been to include all of them and associate a confidence/support score with each gene.

The main focus of Panaroo is to tolerate assembly / annotation errors. While considering contamination or mis-assembly issues, it should be possible to detect and discard those cases in a pre-processing step using popular tools (e.g. CheckM). It's not clear whether pan-genome construction algorithm should be dealing with errors, especially when there are separate tools to do assembly quality-checking and outlier detection. This aspect is not addressed, and justified through experiments.

Regarding simulated experiment: Panaroo shows competitive performance overall, and especially reduces false-positive calls using 'complicated datasets'. Intuition behind these two artificial contamination- and fragmentation- based test cases is not clear to me. For example, in the simulated experiment using contaminated assemblies, why are assemblies not quality-checked before pan-genome construction. Outlier assemblies with very high fragmentation can also be discarded early-on. In addition, please clarify how the highly-fragmented assemblies are being created and what real-scenario do they reflect. Assembly fragmentation often happens at selected sites (e.g., multi-copy genes, repeats) rather than random loci in a genome. What is N50 of fragmented genome assemblies?

Figure 3 reflects false-positive gene clusters present in accessory genome. It is nice to see Panaroo reporting fewer false-positives in the complicated tests. What about false-negatives? A precision-recall curve is needed to better understand the improvements.

A comment on how users should decide between "sensitive" mode and "default" mode of Panaroo is important. I recommend authors to evaluate both modes in all the experiments.

MINOR CONCERNS:

Can Panaroo work with MAGs?

Font size in Figures 2, 4, 5 should be increased for legibility.

We would like to thank both the editor and the two reviewers for taking the time to consider this manuscript in light of recent global events. We realise many researchers are under unusual pressure and very much appreciate the effort.

We have tried to address all the concerns of the reviewers through rewriting, and using new analyses where necessary, and have detailed our responses below. In particular, to our knowledge our approach is the first to use a graphical representation of the pangenome to correct for annotation errors. We have demonstrated the importance of accounting for annotation errors through extensive simulations and the analysis of well understood real world datasets.

Response to reviewers

Reviewer #1

Minor point:

This is by far the most awkward description of what an ortholog might be that I have ever seen: "Orthologs are homologous genes de- scended from the same ancestral sequence in the common ancestor, and not via gene duplication or acquisition."

Quite simply, Orthologs trace their most recent common ancestor to a speciation event. Paralogs trace their most recent common ancestor to a gene duplication event.

We agree completely with the reviewer and have updated the relevant line.

Minor point:

This is a somewhat strange statement: "When analysing bacterial pangenomes we are often interested in not just the function of a gene or protein but also its location, as two nearly identical genes could be under differential regulation at different locations in the genome."

Genes could be regulated differently and still be side-by-side in the genome. Their location and their regulation are orthogonal issues. Location does not necessarily connote regulation.

We have replace this line with

"When analysing bacterial pangenomes we are often interested in identifying paralogs as genes with near identical sequence may perform a different function or be under differential regulation at different locations in the genome."

Minor point:

This is not quite correct: "Many programs for pangenome analysis therefore use location information to further identify paralogs, which occur when two genes descend from the same ancestral sequence due to gene duplication or when a homolog has been acquired horizontally."

When genes are in the same genome as a consequence of HGT, then they are called Xenologs. You can, in fact, have two orthologs in the same genome, if one of them was acquired by HGT. Your statement is conflating a few different issues and so I recommend you straighten out these issues before publication.

We have replaced this line with:

"Many programs for pangenome analysis therefore use location information to further identify paralogs, as well as xenologs, which occur when gene duplications are acquired through horizontal gene transfer."

Minor point:

When you say "Previous approaches for inferring the pangenome include Roary, OrthoMCL, PanOCT, PIRATE, PanX, PGAP, COG-soft, and MultiParanoid (4-10). The majority of methods for determining the pangenome tend to make use of one of two similar approaches (see Supplementary Figure 1)." it is not clear what you mean by "inferring the pangenome" and "determining the pangenome". I suggest you define what you mean in these cases. Do you mean that these approaches can be used in order to help identify all the gene families that are present in a complete pangenome? On its own, a program like OrthoMCL doesn't specifically work on delineating the gene content of a pangenome. If you use it in a particular way, then it can help you carry out this task.

We have added the following line to the start of the introduction to try and clarify what we mean when we say "inferring the pangenome".

"Throughout this paper we refer to the problem of correctly identifying all the gene families that are present in a collection of annotated assemblies as both inferring and determining the pangenome."

We have also updated the mentioned line to:

"Previous approaches which aid in the inference of the pangenome of a collection of bacterial isolates include Roary, OrthoMCL, PanOCT, PIRATE, PanX, PGAP, COGsoft, MultiParanoid, PPanGGoLiN and MetaPGN"

Minor point:

This sentence needs a bit of work: "This can have profound implications for the resulting estimates of the pangenome whereby a higher number genomes leads to a higher number of errors (16, 17)."

Here you speak about "estimates of the pangenome". Again, I believe you are talking about estimating the complete set of gene families in a pangenome, along with their pattern of presence and absence.

We have updated the relevant line to:

"This can have profound implications for the resulting estimates of the the number of gene families present within a pangenome whereby a higher number genomes leads to a higher number of errors"

Reviewer #2

MAJOR CONCERNS:

As many pan-genome construction tools are now available, text in the introduction and results section does not highlight the methodological advance. The use of graph-based representation for characterizing pan-genome is not novel. Pan-genome construction tools like PPanGGOLiN, MetaPGN and PIRATE also use similar graphs where genes are represented as nodes, and edges reflect gene adjacencies. In particular, Panaroo appears very similar to PPanGGOLiN. These tools should not only be discussed, but also included in comparison.

We realise that previous pangenome inference tools do make use of graph based approaches. In fact; Roary, panOCT, PanX and orthoMCL and more recently PPanGGoLin, metaPGN and PIRATE all make use of graphical structures to some extent. However, to our knowledge none of these explicitly attempts to use this structure to correct for annotation errors. To make this clearer, we have replaced the line:

"A subset of these methods then continues by using the neighbourhood or genomic context of each gene to further split orthologous clusters into paralogs."

With

"A subset of these methods then use gene adjacency information to build a graphical representation of the pangenome. This graph is then used to further split orthologous clusters into paralogs. Roary, PIRATE, PPanGGoLin and MetaPGN also provide this graphical representation as an output file."

We have now added PPanGGoLiN to all comparisons in the manuscript. In its default mode we actually found that PPanGGoLiN performed worse than previous methods and that only by enabling the '--defrag' option did its performance compare reasonably with the other tools. Thus we have only included the better performing results in all analyses. Interestingly, PPanGGoLiN does not attempt to split out paralogs and rather reports gene clusters that contain duplicates within a genome. Thus its graphical representation differs significantly in that multiple paralogous genes from the same genome can be clustered together. As one of the main features of most pangenome inference tools is to account for paralogs within the clustering we feel that PPanGGoLiN is in some ways quite divergent to Panaroo and the other tools. One of the main advances of PPanGGoLiN is in its graph partition algorithm in which clusters are grouped into categories such as 'core' and 'accessory' based upon their prevalence and location in the graph. To highlight this we have added the following line to the introduction:

“A final step of some pipelines is to classify the resulting clusters into core and accessory categories based upon their prevalence within the data set. This is usually done using predefined thresholds, however recently model based extensions to this approach have been suggested (Gautreau, 2020).”

After many hours of trying we were unable to get MetaPGN to run successfully. The main MetaPGN pipeline relies on proprietary gene calling software which we were able to install and run. However, in nucleotide mode, after successfully performing the initial clustering of the gene sequences, the pipeline did not generate the final output files. In its amino acid mode, the pipeline hit a memory error in the clustering stage. This was caused by it running cd-hit without adjusting the '-M' flag. After fixing this issue the pipeline still failed to produce the final output files in the amino acid mode.

In addition, graph-based representations have been leveraged before for detecting structural variants, e.g., see Cortex tool (Iqbal et al. 2012).

The structural variation calls provided by Panaroo are only able to identify rearrangements that result in genes being relocated in the genome. However, we believe these gene-level structural variants, when combined with the graphical output provided by Panaroo, allow for additional downstream analyses and simpler interpretation of the resulting calls, as we demonstrate in our analysis of a *N. gonorrhoeae* dataset. To highlight the fact that there are other options available for calling structural variations and in particular finer scale variations we have added the following line to the results:

“Panaroo is only able to call large structural rearrangements that result in genes being relocated within the genome. Finer scale structural variants are better called using assembly graph based approaches such as Cortex (Iqbal, 2012).”

A novel aspect of Panaroo is detection of contamination using poorly supported regions in the graph. However, I did not see a convincing strategy to differentiate errors vs. rare genes or recombinations. From a practical perspective, it's unclear whether discarding poorly supported regions is a good idea as it also affects sensitivity of the method. Perhaps a better way could have been to include all of them and associate a confidence/support score with each gene.

Panaroo will not remove rare genes if they are embedded within the main pangenome graph and thus have sufficient contextual support. Thus a gene cluster can be present in only a single genome and remain in the graph even in the strictest mode. We realise that we only describe this fact in the methods and thus have added the following line to the results to clarify that rare genes with contextual support are kept.

“Potentially contaminating genes with low contextual support in the graph are then optionally removed. This retains rare genes that have reliable contextual support.”

We agree that there is always a trade-off to be made in removing errors possibly at the expense of eliminating real genes. Panaroo includes a number of different thresholds or ‘scores’ that can be changed by the user depending upon their preference and research question. In many situations including all gene clusters can result in a dataset where the number of erroneous clusters can dominate the downstream analyses. In these situations removing errors is essential. We felt that generating a meta score would be hard for users to interpret and thus we defined three modes. In Panaroo’s ‘sensitive’ mode it does not remove any gene clusters. The resulting Panaroo graph file also includes the support for each cluster and thus running Panaroo in its ‘sensitive’ mode corresponds fairly well with the request above. To better highlight this fact and to help users decide which mode to use we have run Panaroo in sensitive mode in all comparisons and added the following text to the results

“Panaroo uses a number of predefined thresholds to construct the pangenome graph. These can all be adjusted by the user but we provide a number of modes for common use cases. In the ‘strict’ mode Panaroo takes a more aggressive approach to contamination and erroneous annotation removal. This is most useful when investigating genomes where rare plasmids are not expected or when phylogenetic parameters such as gene gain and loss rates are of interest. In these cases erroneous gene clusters can quickly dominate the estimated parameters. In its ‘sensitive’ mode Panaroo does not remove any gene clusters. This is useful if a researcher is interested in rare plasmids which may be hard to distinguish from contamination. When running Panaroo in sensitive mode it is important to be aware of the possibility of a higher number of erroneous clusters. In the following analyses we run Panaroo in both its ‘strict’ and ‘sensitive’ modes with Panaroo generally outperforming all other tools even without contamination removal.”

The main focus of Panaroo is to tolerate assembly / annotation errors. While considering contamination or mis-assembly issues, it should be possible to detect and discard those cases in a pre-processing step using popular tools (e.g, CheckM). It's not clear whether the pan-genome construction algorithm should be dealing with errors, especially when there are separate tools to do assembly quality-checking and outlier detection. This aspect is not addressed, and justified through experiments.

We agree that initial quality control is an important step prior to running pangenome inference methods which is why we provide and describe the 'panaroo-qc' script which produces a number of diagnostic plots to allow a researcher to remove very poor assemblies. However, discarding all genomes that have any form of misannotation, fragmentation or contamination can result in large reductions in the number of usable genomes. Additionally, even very low annotation error rates will compound to produce many erroneous clusters in large datasets.

To better discuss this issue in the manuscript we ran CheckM on the Mtb dataset. This indicated that a subset of samples had a higher number of gene calls and lower genome completeness. As we know that this dataset should be highly conserved these genomes could be flagged as containing a higher number of annotation errors. In a more diverse species such as *Klebsiella pneumoniae* the small difference observed here would be indistinguishable from plausible variation. Additionally, by using Panaroo rather than discarding the genomes we prevented a data loss of 12% of genomes. This could conceivably add considerable power to downstream analyses. We have added the following text to the results as well as supplementary figures to the supplementary materials to illustrate the results of CheckM.

“An alternative to correcting gene annotations is to perform strict quality control checks on assemblies prior to running pangenome inference tools. For very highly contaminated assemblies or those of very low quality this can be the best option and Panaroo includes a quality control pre-processing script for this purpose. However, in many cases low level contamination and fragmented assemblies are common and thus filtering out assemblies with minor errors can lead to large data losses. In large collections even very low annotation error rates will eventually compound pangenome inference results. To investigate such a strategy on the Mtb dataset we ran CheckM a common assembly quality control pipeline (Parks, 2015). CheckM produces completeness and contamination scores by using a reference gene data set to compare with assemblies. The resulting scores on the Mtb dataset are given in Supplementary Figure 2. As we know this data set should contain highly similar assemblies it is possible to identify a number of problematic genomes with slightly lower completeness scores. If we were to remove these genomes it would result in a loss of 12% of the data set which could potentially have a large impact on downstream analyses. Instead, using Panaroo we are able to retain these assemblies whilst controlling the error rate.”

Regarding simulated experiment: Panaroo shows competitive performance overall, and especially reduces false-positive calls using 'complicated datasets'. Intuition behind

these two artificial contamination- and fragmentation- based test cases is not clear to me. For example, in the simulated experiment using contaminated assemblies, why are assemblies not quality-checked before pan-genome construction. Outlier assemblies with very high fragmentation can also be discarded early-on. In addition, please clarify how the highly-fragmented assemblies are being created and what real-scenario do they reflect. Assembly fragmentation often happens at selected sites (e.g., multi-copy genes, repeats) rather than random loci in a genome. What is N50 of fragmented genome assemblies?

For the reasons described above we believe that simply discarding any assembly with an error is not a desirable solution as it can result in large data losses. The ‘complicated’ datasets were included to demonstrate the large impact that these sources of error could have on pangenome inference tools. Contamination and fragmentation are well known issues in the annotation of de novo microbial assemblies with a number of previous papers describing their impact [1,2]. We feel that the demonstration of the well understood Mtb dataset further indicates that annotation errors do have a profound impact in a real world example. We realise that no simulation is perfect and recognise that the ‘complicated’ simulations contain more errors than might be expected in a normal dataset. However, we feel that the simple simulations with low error rates provide a suitable comparison in a low error rate scenario and indicate that Panaroo still outperforms previous tools. To clarify that the ‘complicated’ simulations represent unusually high error rates we have adjusted a portion of the text in the results to read:

“In addition, we simulated two more complicated datasets, one of which had an increased level of fragmentation of the assembly by fragmenting the input genome prior to the NGS simulation. This resulted in a mean N50 of 23,112. The second more complex simulation included contamination by randomly adding in short fragments of the Staphylococcus epidermidis reference genome, which is a common contaminant. The more complicated simulations represent data sets with unusually high error rates but help to clarify the large impact that these sources of error can have on pangenome inference as was previously demonstrated in the analysis of the highly clonal Mtb data set.”

Figure 3 reflects false-positive gene clusters present in the accessory genome. It is nice to see Panaroo reporting fewer false-positives in the complicated tests. What about false-negatives? A precision-recall curve is needed to better understand the improvements.

Figure 3 does include false negatives although we realise our description was not very clear. The ‘missing genes’ refers to simulated annotations that were not found in the resulting pangenome inference. These are false negatives and are included in the ‘total error’ count given in Figure 3a. We have improved the annotation of the figure and added the following line to the caption to better illustrate this point.

“Missing genes refer to false negative gene calls where the annotation is not present in the final pangenome.”

Precision-recall curves are usually used to assess classification algorithms rather than the clustering problem we are considering. As most of the algorithms including Panaroo do not have a single threshold which can be scaled it is not possible to generate a precision-recall curve to compare them. A similar and common alternative for clustering problems is to look at the rand index which presents a single summary measure of the similarity of two clusterings. It considers whether gene pairs appear together in both the simulated and inferred clusters. As there is a large core genome which all methods correctly cluster for the most part this dominates the rand index. Consequently, all methods achieve a rand score greater than 0.99998 out a maximum of 1. This makes the interpretation of the rand index difficult and thus we opted to describe the total number of the different sources of error.

A comment on how users should decide between "sensitive" mode and "default" mode of Panaroo is important. I recommend authors to evaluate both modes in all the experiments.

Please see the response discussing the different modes of Panaroo above.

MINOR CONCERNS:

Can Panaroo work with MAGs?

Panaroo is not recommended for metagenomic datasets. We have added the following line to the introduction to reflect this.

“Whilst these scripts can allow for comparisons of the resulting pangenomes between species Panaroo is not recommended for metagenomic datasets.”

References

1. Denton JF, Lugo-Martinez J, Tucker AE, Schrider DR, Warren WC, Hahn MW. Extensive error in the number of genes inferred from draft genome assemblies. *PLoS Comput Biol.* 2014;10:e1003998.
2. Salzberg SL. Next-generation genome annotation: we still struggle to get it right. *Genome Biol.* 2019;20:92.